# Fabrication of Chitosan/Hydroxyethyl Cellulose/TiO_2_ Incorporated Mulberry Anthocyanin 3D-Printed Bilayer Films for Quality of Litchis

**DOI:** 10.3390/foods11203286

**Published:** 2022-10-20

**Authors:** Jinjie Luo, Guofeng Xia, Lizi Liu, Anping Ji, Qiang Luo

**Affiliations:** Department of Mechanical Engineeering, Chongqing Three Gorges University, Chongqing 404000, China

**Keywords:** mulberry anthocyanins, 3D printing, litchi

## Abstract

In this study, a bilayer antibacterial chromogenic material was prepared using chitosan (CS) and hydroxyethyl cellulose (HEC) as inner substrate, mulberry anthocyanins (MA) as a natural tracer, and titanium dioxide nanoparticles (nano-TiO_2_)/CS:HEC as a bacteriostatic agent for the outer layer. By investigating their apparent viscosity and suitability for 3D printing links, the optimal ratio of the substrates was determined to be CS:HEC = 3:3. Viscosity of the CH was moderate. The printing process was consistent and exhibited no breakage or clogging. The printed image was highly stable and not susceptible to collapse and diffusion. Scanning electron microscopy and infrared spectroscopy indicated that intermolecular binding between the substances exhibited good compatibility. Titanium dioxide nanoparticles (nano-TiO_2_) were evenly distributed in the CH and no agglomeration was observed. The inner film fill rates affected the overall performance of the chromogenic material, with strong inhibitory effects against Escherichia coli and Staphylococcus aureus at different temperatures, as well as strong color stability. The experimental results indicated that the double-layer antibacterial chromogenic material can, to a certain extent, extend the shelf life of litchi fruit and determine the extent of its freshness. Therefore, from this study, we can infer that the research and development of active materials have a certain reference value.

## 1. Introduction

Litchi (*Litchi chinensis Sonn.*) is an evergreen fruit crop of the subtropical region that has a high commercial value because of its pleasant flavored translucent aril, refreshing taste, and nutritive value [1]. Consequently, post-harvest handling has become a key concern in the development of litchi production [2]. The attractive, red-colored pericarp of litchi rapidly turns brown within 1–2 d and loses its commercial value in addition to consumer acceptance within this period [3]; thus, under ambient conditions, the fruit has a significantly limited shelf life of 2–3 d [4,5]. Therefore, there are difficulties associated with the storage, transportation, and marketing of the harvested litchi fruit, resulting in large economic losses due to the limited preservation technology associated with this fruit [6,7,8]. There is increasing public concern regarding the food safety of litchi, which justifies the need for novel, safe, effective, and environmentally friendly methods to control post-harvest pathogenic diseases of litchi fruit [9].

The food packaging sector plays a major role in the global plastic market. With recent developments in science and technology, green, pollution-free, and degradable packaging materials have attracted increasing research attention [10]. However, because of consumer preference for minimally processed and natural products, active packaging has gained importance. The new goals in the research and development of packaging materials are to improve the functionality of food packaging materials based on degradability, provide more comprehensive protection for food, and allow consumers to experience the convenience brought about by more intuitive packaging [11]. Intelligent packaging materials usually consist of indicators that react with specific volatile substances produced during the food storage process to achieve real-time monitoring of food quality [12,13]. Anthocyanins are naturally found in fruits and petals and can produce color reactions under varying pH conditions [14]. In recent years, they have been used as natural indicators and are included in intelligent packaging materials for the real-time monitoring of food quality [15].

In recent years, multilayer materials have been widely used in the food industry because of their excellent performance. Electrospinning and other technologies are often used to develop materials that retain the freshness of food [16,17]. Three-dimensional (3D) printing is an extremely convenient emerging technology that exhibits unique advantages in the preparation of patterned functional materials [18,19]. Further, 3D printing has many notable characteristics, such as fast production speed, short cycle time, low cost and pollution, in addition to high accuracy, material utilization, and adaptability. Thermal fusion deposition and cold forming technologies require relatively low material types and properties and have a wide range of viscosities. The 3D-printing technique used for printing fluid substances, such as food and gels, is based on these technologies. However, very few studies have been conducted on the use of 3D-printing technology to prepare customized packaging materials. This has limited the development and application of 3D-printing technology in the field of food packaging [20]. Based on the above-mentioned background, the development of a biodegradable, pattern-based, intelligent packaging material, and its application in the preservation of litchi fruit and intelligent color display are in line with the developing trend of contemporary food packaging.

Mulberry anthocyanin (MA) is an edible natural pigment with non-toxic, water-soluble, age-delaying, and antioxidant properties. MAs exhibit the same properties as many other anthocyanins. They also undergo structural changes when the environmental pH is changed, resulting in different colors. Therefore, they can be used as pH-sensitive indicators in the preparation of color-indicating smart packaging materials, which have large development potential. Chitosan (CS) has many excellent properties, such as biocompatibility, film formation, selective permeability, barrier, toughness, and antibacterial and antioxidant properties [21]. In addition, CS is a safe, non-toxic, and edible animal fiber. In addition, CS is environmentally friendly and biodegradable and is widely used in the chemical, food, and biomedical fields [22,23]. It is also highly valued in the development and preparation of food packaging materials, especially active and smart packaging materials [24,25]. Hydroxyethyl cellulose has excellent properties, such as biocompatibility, film formation, and biodegradability. It was widely used by researchers in many applications, such as coating preparation, biopharmaceuticals, polymerization, catalysts, sensors, and textiles [26]. Nano titanium dioxide is a white, loose powder that acts as a UV shielding agent and provides protection against UV light to some extent. It has excellent properties such as dispersibility, tolerance, barrier, biodegradability, bacterial inhibition, and self-cleaning, which make this material relatively stable. Many studies were conducted to apply packaging materials made of nano-TiO_2_ to the preservation of fruits and vegetables; these studies revealed that the addition of nano-TiO_2_ could not only improve the mechanical and antibacterial properties of the materials but also effectively extend the shelf life of food [27].

In this study, 3D-printing technology was used to prepare bilayer films with CS and hydroxyethyl cellulose as the substrate. Natural indicator MA was added to the inner material to provide antibacterial properties and freshness indication, and the outer layer was modified with nano-TiO_2_ to enhance the antibacterial properties of the packaging. Next, the performance of bilayer films to monitor the freshness of litchi was investigated. The application of this material in the field of food packaging could provide a reference for the application of 3D-printing technology in the field of food packaging.

## 2. Materials and Methods

### 2.1. Materials

Mulberry anthocyanin (MA) was purchased from Shaanxi Cuiyajia Biotechnology Co., Ltd. (Xian, China) Chitosan (CS, deacetylation degree >90%) was purchased from Zhuhai Taste Good Food Additives Co., Ltd. (Zhuhai, China). Hydroxyethyl cellulose (HEC viscosity: 100,000) was purchased from Qingdao Maoshi International Trading Co., Ltd. (Qingdao, China). Nano-TiO_2_ (average particle size: 10 nm) was purchased from Hebei Maisen Titanium Dioxide Co., Ltd. (Shijiazhuang, China). Litchi (Litchi chinensis Sonn. cv. Feizixiao) was supplied by the Guoguoxiang Manyuan ecological family farm. Unless otherwise stated, all the other chemicals and solvents used were of reagent grade and were purchased from Chongqing Sinopharm Chemical Reagent Co. (Chongqing, China).

### 2.2. Preparation of Chitosan/Hydroxyethyl Cellulose Substrate

A 2% acetic acid solution was prepared. Subsequently, 6% (*w*/*v*) CS/HEC powder was added to the acetic acid solution, followed by stirring in a constant temperature water bath at 37 °C until the powder was completely dissolved. The mixture was sonicated for 60 min to remove any bubbles. Five different film substrates were prepared using different proportions of the constituents in the CS/HEC mixed powder (1:5, 2:4, 3:3, 4:2, and 5:1). The abbreviation of materials used for the film substrate is shown in Table 1.

### 2.3. Preparation of Dual-Layer Antimicrobial Color Development Materials

The outer film-forming solution was prepared by accurately weighing 1% (*w*/*v*) nano-TiO_2_ and adding it to the pre-configured CS/HEC substrate solution. This solution was stirred at room temperature until nano-TiO_2_ was uniformly dispersed in the CS/HEC substrate solution. Subsequently, nano-TiO_2_ was removed by ultrasonic de-bubbling for 60 min and left aside. The inner film-forming solution was prepared by accurately weighing 0.5% (*w*/*v*) MA and adding it to the pre-configured CS/HEC substrate solution. The solution was stirred at room temperature until MA was completely dissolved in the CS/HEC substrate solution. The solution was removed after a uniform color was obtained. Subsequently, the bubbles were removed by ultrasonication for 60 min, after which the solution was left aside. The outer film was prepared by slowly and evenly pouring the outer film solution (20 mL) onto a glass plate with dimensions of 15 cm × 15 cm, which was then placed in an oven at 45 °C for 30 min. The glass plate was removed when the surface of the outer film reached a solid and viscous semi-dry state. The glass plate was then left aside. The procedural steps for the preparation of the double-layer antibacterial chromogenic material are shown in Figure 1. The outer layer film (in a semi-dry state) was placed in the printing area of the 3D printer (3DP; PR-D3, Sichuan Foster company, (Chengdu, China), which was filled with the inner-layer film-forming solution. The instrument was run, and a bottom-removing single layer was printed. The first film was removed after printing, dried in an oven at 30 °C for 30 min, and subsequently removed and dried naturally for 24 h. The films were then stored. Five bilayer, antimicrobial, and chromogenic materials were obtained by varying the filling rate of the inner layer (0%, 15%, 30%, 45%, and 60%).

### 2.4. Scanning Electron Microscopy

The surfaces of the CHT and CHM were scanned using scanning electron microscopy (SEM; Hitachi SUI510, Hitachi Ltd., Tokyo, Japan) to observe the compatibility between the materials and the dispersion of the nano-TiO_2_ in the materials.

### 2.5. Fourier-Transform Infrared Spectroscopy

Fourier-transform infrared (FTIR) spectroscopy (FT-IR NICOLET iS10, Thermo Fisher, Waltham, China) was subsequently used to analyze the changes in the chemical structure of each material between wavenumbers 4000 cm^−1^ and 500 cm^−1^ at a resolution of 4 cm^−1^, and the scanning rate was maintained at 32 s^−1^.

### 2.6. Thickness and Water Vapor Transmission Coefficient

The thickness of each material was measured using a micrometer thickness gauge. Four points were randomly selected around the material, and one point was randomly selected in the middle, that is, five points were measured. Each group of films was measured five times, and the thickness of the film was obtained after averaging the values obtained. The water permeabilities of the five materials were measured according to the method proposed by Fabra et al. [28]. The silica gel was placed in a drying tower (25 °C, 0% relative humidity). Water was added to the test tube and five materials were used for sealing. The control group was sealed with aluminum foil to calculate the solution loss during the sealing process. The test tube weight was measured every 2 h with an analytical balance (accuracy of 0.0001 g), and the water vapor transmission coefficient of the material was obtained by calculating the slope of the weightlessness curve. The mass loss was obtained by subtracting the sealing loss from the total mass loss.

### 2.7. Mechanical Properties

The area with the uniform and flat material was selected and cut into a rectangular sample with dimensions of 80 mm × 10 mm. The tensile strength (TS) and elongation at break (E) of the material were measured using a tensile testing machine (HDB609B-S, Haida International Instrument Equipment Co., Ltd., Taichung, China) to evaluate the mechanical properties of the material [29]. Eight parallel samples were cut from each material and the average value was calculated.

### 2.8. Opacity

The five materials were cut into rectangular samples with dimensions of 10 mm × 40 mm, placed in a 10 mm colorimetric dish, and fixed on the inner wall of the dish. The light transmittance of the sample was measured at a wavelength of 600 nm using a UV-visible spectrophotometer (PerkinElmer, Waltham, MA, USA) [30]. Each group of samples was measured six times in parallel, and the average value was calculated.

### 2.9. Broad-Spectrum Antibacterial

The antibacterial properties of CH, CHT, and CHM against Gram-positive S. aureus and Gram-negative E. coli were measured under natural and UV light, respectively [31]. This was performed to determine the influence of different light conditions on the broad-spectrum antibacterial properties of the three materials and to identify the main sources of the antibacterial effect of the double-layer antibacterial chromogenic materials. The sizes of the antibacterial circles of the five materials were measured with reference to the technology proposed by Otini et al. [32]. The inner-layer film was placed face down in direct contact with bacteria to evaluate the influence of the printing filling rate (of the inner-layer film) on the broad-spectrum antibacterial effect of the double-layer antibacterial colorimetry material.

### 2.10. Color Stability

The chromatic aberration method was used to assess the color stability of the five materials [33]. These materials were placed in a chamber at a constant temperature (4 °C, 25 °C, and 37 °C) and humidity (60%). The *L**, *a* *, and *b* * values of the materials were measured every 2 d for a total of six times, and three groups of data were measured each time. The total rate of change of *L**, *a**, and *b** was calculated to characterize the color stability of the material. The specific calculation formula is as follows:(1)$$\Delta E=\sqrt{{\left(\Delta L\right)}^{2}+{\left(\Delta a\right)}^{2}+{\left(\Delta b\right)}^{2}}$$
where Δ*E* is the total color difference of the material; Δ*L* = *L** − *L*0*; Δ*a* = *a** − *a*0*; Δ*b* = *b** − *b*0*; *L**, *a**, and *b** are the measured values after film placement; and *L*0*, *a*0*, and *b0** are the initial values measured before film placement.

### 2.11. Preservation of Litchi Fruit

The preservation and color effects of the double-layer, antimicrobial, and color development materials on the litchi fruits were studied at 25 °C and 4 °C. The litchi fruits were placed in 500 mL PE containers (approximately 14 cm × 9 cm × 4 cm in length, width, and height, respectively), with six fruits in each box. The double-layer, antimicrobial, color development material was cut into rectangular samples with dimensions of 15 cm × 10 cm. The litchi in the control group was not sealed. They were placed in an environment with a relative humidity of 85–95% for the preservation experiments.

### 2.12. Sensory Evaluation

The sensory evaluation group was established with 10 members. Combining the real-time conditions of various samples, the sensory evaluation was conducted on the following four parameters: (a) peel state, (b) pulp state, (c) fruit odor, and (d) pulp taste. The highest and lowest awardable scores were 100 and 0, respectively. The specific scoring parameters and evaluation criteria are listed in Table 2. The litchi stored at 25 °C was measured every 2 d up to 8 d, while the litchi stored at 4 °C was measured every 10 d up to 40 d.

### 2.13. Browning Index

The litchi stored at 25 °C was inspected every 1 day for a total of 8 day, and the litchi stored at 4 °C was inspected every 5 day for a total of 40 day. The browning grade was evaluated by the proportion of browning area on the surface of the litchi to the total area of the litchi pericarp. The following grades were awarded based on browning [34]. First, second, third, fourth and fifth class each meant no browning on the surface, minimal browning on the surface, browning area accounts for <1/4 of the total area, browning area accounts for 1/4–1/2 of the total area, and browning area accounts for >1/2 of the total area, respectively. The browning index was calculated using the following formula:(2)$$Browning\;index\;of\;litchi=\frac{\sum^{}(\mathrm{Browning}\;\mathrm{level}\times \mathrm{The}\;\mathrm{number}\;\mathrm{of}\;\mathrm{litchi}\;\mathrm{in}\;\mathrm{this}\;\mathrm{class})}{\mathrm{Browning}\;\mathrm{peak}\times \mathrm{The}\;\mathrm{total}\;\mathrm{number}\;\mathrm{of}\;\mathrm{litchi}}$$

### 2.14. Determination of Soluble Solids Content

The content of soluble solids in litchi during storage was determined using a hand-held saccharometer [35]. The litchi pulp was extracted and vigorously crushed in a mortar. Subsequently, a drop of juice was placed on the prism glass of the saccharometer, which lightly covered the protective lid, thereby allowing the juice to evenly spread all over the glass. Thereafter, the sample was observed and the reading was recorded. The litchi stored at 25 °C was observed every 1 day for 8 day, and the litchi stored at 4 °C was observed every 5 day for 40 day.

### 2.15. Determination of Titratable Acid Content

The change in titratable acid content in litchi fruit during storage was determined by acid-base neutralization titration. Initially, the litchi pulp was removed, placed in a mortar, and completely crushed. The crushed litchi pulp (10 g) and distilled water (20 mL) were added to a beaker, stirred well, and subsequently placed in a water bath for heating at 75 °C. After 30 min, the contents of the beaker were removed and placed in a 100 mL volumetric flask for constant volume measurement. After filtration, two drops of phenolphthalein were added to 20 mL of the filtrate and titrated with NaOH. The litchi stored at 25 °C was titrated every 1 day for 8 day, and the litchi stored at 4 °C was titrated every 5 day for 40 day.

### 2.16. Evaluation of the Intelligent Color

The litchi fruits were placed in PE containers, sealed with a double layer of antibacterial color material, and placed in two constant temperature environments (4 °C and 25 °C). The chromaticity value of the material was measured every two days to evaluate its intelligent color-rendering effect, and the result was expressed as the total chromaticity change value, that is, Δ*E*.

### 2.17. Statistical Analysis

SPSS 24.0 software was used to process the data obtained in the experiment, and Duncan’s multiple comparative test method was used for significance analysis, in which *p* < 0.05 indicated that the data had significant differences. Data processing software such as Origin 2017 was used to analyze the experimental data and draw relevant charts.

## 3. Results and Discussion

### 3.1. Performance Testing of the Dual-Layer Antimicrobial Color

The compatibility between the materials can be analyzed through SEM plots by evaluating the roughness of the material surfaces. A greater surface roughness indicates low compatibility, whereas flatter and smoother surfaces indicate greater compatibility between the materials [36]. The apparent morphology and SEM images of CHM and CHT are shown in Figure 2. It is clear from the apparent morphology that both CHM and CHT have a flat and smooth appearance in addition to uniform texture and color, without any breakage, bubbles, and holes. The CHM presents a smooth, flat, and uniform surface without producing any fractures, micropores, or protrusions, which indicates good compatibility between each material added to CHM. MA was uniformly dispersed in the CH matrix and exhibited a slightly rougher surface than CHM, even though it presented a completely dense image. The bacterial inhibitory effect of the materials increased as the concentration of the nanoparticles increased. However, when the concentration of the nanoparticles was significantly high, agglomeration occurred in the matrix, leading to adverse effects on its various properties. From the SEM image, it can be observed that the nano-TiO_2_ particles in the CHT are more dispersed in the matrix and no aggregation occurs, which indicates that the addition of nano-TiO_2_ in CHT is appropriate. This indicates a positive effect on the water vapor transmission coefficient and the mechanical properties of the material [16].

### 3.2. Fourier-Transform Infrared Spectroscopy

In this experiment, the infrared spectra of various added materials, that is, CH, CHT, and CHM, were analyzed. The results are shown in Figure 2, where the characteristic absorption peak of nano-TiO_2_ was approximately 693 cm^−1^. The absorption peak at 1633 cm^−1^ was generated by the C=C vibration of the aromatic ring frame in the MA aromatic substance, and the HEC peak at 2865 cm^−1^ was attributed to the C–H aliphatic stretching vibrations. The characteristic band at 3386 cm^−1^ was attributed to the O–H stretching vibrations. The peak at 1150 cm^−1^ corresponded to the secondary alcohol group of HEC. In the infrared spectrum of CS, a broad absorption peak at 3800–3000 cm^−1^ was observed. The broad and strong absorption peaks observed at 3800–3000 cm^−1^ were the superimposed effects of the absorption peaks of –OH and –NH_2_ stretching vibrations, and the two bands observed at 2920 cm^−1^ and 2811 cm^−1^ represent the presence of C–H aliphatic stretching vibrations. There was a shifting of the wide peak between 3800–3000 cm^−1^, which suggested hydrogen bonding between polymer components including an interaction between polymers and TiO_2_ via hydrogen bonding [37]. In CH, CHT, and CHM, the broad peaks at 3000–2750 cm^−1^ indicate the presence of –CH stretching vibrations in the material and the absorption peaks at 1750–1500 cm^−1^ provided information on not only the bending vibrations of water molecules but also on the symmetric stretching vibrations of –COO. In CS and CH, the C–O stretching vibration absorption peaks were 1091 cm^−1^ and 1047 cm^−1^, respectively. The corresponding absorption peaks in CHT and CHM shifted to 1027 cm^−1^ and 1038 cm^−1^, respectively, indicating the existence of intermolecular interaction forces between the materials.

### 3.3. Optical Properties and Appearance

The appearance and shape of the material also greatly influence the consumer’s choice of the product. The color of the material was characterized using a colorimeter by evaluating the *L**, *a**, *b**, and ΔE values. The area where the bilayer intersected was chosen for the experiment. As shown in Table 3, the colors of the five materials varied, with CHMT-0 being the brightest of the five materials, and the brightness of each material significantly decreased (*p <* 0.05) as the print fill rate increased. This was due to the addition of anthocyanins in the inner-layer material. It was observed that as the print fill rate increased, the dark inner-layer material gradually covered the translucent outer layer material, which resulted in an increase in the *L** and *a** values and a decrease in the *b** value of the material. The brightness of the double-layer antimicrobial chromogenic material gradually decreased as the print fill rate of the inner-layer film increased and the overall color exhibited a blue–purple shade. As observed in Table 3, the opacity of CHMT-0 was the lowest among the five materials, and the opacity of the material gradually increased with an increase in the filling rate of the inner film printing; the material opacity was the highest when the filling rate of the inner material reached 60% (CHMT-60%).

### 3.4. Water Vapor Transmission Coefficient

The water vapor transmission coefficient of packaging materials should generally be as small as possible to prevent accelerated spoilage and deterioration of food products caused by massive water loss during storage. In this experiment, the water vapor permeability of the five prepared double-layered antimicrobial coloring materials was tested to characterize the water vapor barrier performance of each material, and the test results are presented in Table 4. It can be observed that the thickness of the five materials increased with an increase in the print filling rate, among which, the thickness of CHMT-0 was the smallest, and the thickness of the other four double-layer materials increased by 58–117.9% compared to CHMT-0. The water vapor transmission coefficient of CHMT-0 was (7.81 ± 0.25) × 10^–12^ g·cm‧(cm^2^·s·Pa)^−1^, while the water vapor transmission coefficient of the double-layered antimicrobial chromogenic material gradually decreased with an increase in the print fill rate. The water vapor transmission of the double-layered material was observed to be low ((2.16 ± 0.15 × 10^–12^ g·cm‧(cm^2^·s·Pa)^−1^), which indicated that this material had the strongest water vapor barrier. Denser matrices with few pores or void spaces lead to lower permeability of the polymeric matrices [38]. As the print fill rate increased, the original number of print pores on the inner film gradually decreased. As the double-layered structure of the material became denser and the thickness of the material increased, more and more water molecules passed through the two layers in succession and diffused outward. Hence, the overall water resistance of the material significantly improved (*p <* 0.05).

### 3.5. Mechanical Properties

The mechanical properties of packaging materials largely determine the safety of food products. The mechanical properties of packaging materials are commonly characterized by their TS and E [39]. The mechanical properties of the five double-layered antimicrobial coloring materials are listed in Table 2. It can be observed that TS increases with an increase in the print fill rate of the inner film. Notably, when the print fill rate of the inner film was 0% (CHMT-0), the TS of the material was 21.55 ± 0.58 MPa, which was the lowest among the five materials. The TS of the material gradually increased as the print fill rate of the inner film increased, and when the print fill rate was 60% (CHMT-60%), the TS of the material was the highest, reaching a value of 34.43 ± 1.12 MPa, with an annual increase of 59.8%. The E value usually reflects the toughness of the material, and its lowest value was 17.01 ± 0.11% when the print fill rate was 0% (CHMT-0). Moreover, the elongation at the break of the material gradually increased with an increase in the fill rate and thickness. When the printing fill rate reached 60% (CHMT-60%), the E of the material reached 24.01 ± 0.73%, which was 41.2% higher than that of CHMT-0. The filling improved the homogeneity of the polymeric matrices, possibly by filling up the void spaces, giving a better distribution of stress due to applied force and improved mechanical properties [40].

### 3.6. Broad-Spectrum Antibacterial

In this experiment, the broad-spectrum bacterial inhibition of CH, CHT, and CHM was tested to determine the main sources of broad-spectrum bacterial inhibition of the bilayer antimicrobial chromogenic materials (Figure 3). This was carried out by selecting *E. coli* as a representative species of Gram-negative bacteria and *S**. aureus* as a representative species of Gram-positive bacteria. In addition, the three materials, CH, CHT, and CHM, were treated with daylight and UV light (25 W) prior to the experiment to investigate the changes in the broad-spectrum bacterial inhibitory properties after treatment with varying light conditions. Figure 4 depicts the inhibition rates of CH, CHT, and CHM against *E. coli* and *S. aureus* after treatment with varying light conditions, where the materials were first treated under UV light (25 W) for 24 h prior to the experiment in the UV light group. It can be observed from Figure 4 that the CH substrate exhibited some antibacterial properties and the inhibition rates against *E. coli* and *S. aureus* were 32.13 ± 1.74% and 36.51% ± 1.43%, respectively, under daylight conditions, because the CS added to the material could electrostatically interact with the cell walls of the Gram-positive and Gram-negative bacteria, or with anionic lipopolysaccharides, leading to eventual cell death. The bactericidal effect of CH on Gram-positive bacteria was stronger than that on Gram-negative bacteria. UV illumination had a minimal effect on CH inhibition. The addition of nano-TiO_2_ led to a substantial improvement in the broad-spectrum bacterial activity of CHT, which reached 67.42 ± 0.92% and 70.44 ± 2.01% against E. coli and S. aureus, respectively, under daylight conditions. The bacterial inhibition of CHT was significantly enhanced after the UV illumination (25 W) treatment, reaching 89.56 ± 1.21% and 91.44 ± 1.98% against E. coli and S. aureus, respectively. This was due to the stronger oxidation ability of nano-TiO_2_ upon UV irradiation, which could lead to cell rupture of microorganisms and exhibit stronger antibacterial activity [41]. The addition of MA led to the broad-spectrum bacterial inhibition of CHM, which was slightly higher than that of CH after treatment with both daylight conditions and UV light (25 W). Studies have reported that anthocyanins themselves have a certain bacterial inhibition effect, the strength of which can be related to the structure and nature of the anthocyanins. However, the data from this experiment indicated that MA had a weaker bacterial inhibition performance. Under daylight conditions, the inhibition rates of CHM against *E. coli* and *S. aureus* were 41.55 ± 0.79% and 42.08 ± 1.67%, respectively, while after the UV light (25 W) treatment, the inhibition rates of CHM against *E. coli* and *S. aureus* decreased to 34.78 ± 0.75% and 38.40 ± 1.64%, respectively. This may be due to the acceleration of MA decomposition by UV light irradiation, while the inhibition rate remains higher than the inhibition rate of CH, indicating that CS and HEC play a protective role against MA [42]. In summary, the broad-spectrum bacterial inhibition of the materials is primarily caused by the addition of CS and nano-TiO_2_ to the substrate and outer film, respectively. Bacterial inhibition of the outer material was significantly higher than the bacterial inhibition of the inner material. Combined with the above experiments, the inhibition circle method was used to test the inhibition performance of five types of double-layer antibacterial coloring materials, CHMT-0, CHMT-15%, CHMT-30%, CHMT-45%, and CHMT-60%; the inner layer of the material was facing down and in direct contact with the bacteria, simulating the real-time packaging process in which the material is in direct contact with the food surface. The test results are presented in Table 5. The UV illumination group was treated with UV (25 W) for 24 h prior to the experiment. This result indicates that the filling rate is significantly high, which may lead to a decrease in the bacterial inhibition effect of the bilayer antimicrobial chromogenic material.

### 3.7. Color Stability of the Double-Layer Antibacterial Color Development Material

Consumers judge freshness and other information regarding packaged goods by identifying the color of intelligent color-rendering materials [43,44]. However, the color of the materials may change under the influence of the external environment, which affects the color-rendering results. Therefore, it is necessary to explore the color stability of these materials [45].

In this experiment, the color stability of each material was studied at three different temperatures, and the results were expressed as the total color difference change value ΔE. A smaller change in color difference indicates that the material is more stable during storage. When ΔE < 3.5, it can be inferred that the change in color cannot be perceived by the naked eye. As shown in Figure 5, all six materials underwent varying degrees of color change during storage. The total color variation value increased gradually with storage time, and the degree of change decreased with decreasing temperature (37 °C > 25 °C > 4 °C). This was caused by the reaction of the diphenylchroman cation in the anthocyanins, which were transformed into chalcone and pseudobase and caused the overall color change of the materials. The degradation rate of the anthocyanins was accelerated at high temperatures, and the rate of change of color was further accelerated. CHM was the least stable at all three storage temperatures, which indicated that the outer material could provide some protection to the anthocyanins added to the inner material. Except for the two materials mentioned above, the stability of the remaining four bilayer materials increased with increasing print fill rate, and CHMT-60% exhibited optimal stability with ΔE values (12th day of storage) of 2.14 ± 0.08, 5.14 ± 0.15, and 17.32 ± 0.42 at 4 °C, 25 °C, and 37 °C, respectively. With increasing print fill rate, the anthocyanin added to the inner-layer material gradually decreased the specific surface area, while the chitosan/hydroxyethyl cellulose matrix played a protective role against anthocyanins, which, in turn, led to an improvement in anthocyanin stability [42]. The comparative chart of the litchi storage period is shown in Figure 5. The relative humidity of the litchi storage environment was approximately 85–95%. It can be observed from Figure 5 that the litchi fruit in the control group experienced serious water loss after storage compared to the experimental group. The reduction in fruit volume was higher and the area of the mold was larger. From the above results, it can be noted that the double-layer antibacterial chromogenic material can effectively delay water loss from litchi fruit, and to a certain extent, inhibit litchi spoilage. According to the above experimental results, CHMT-45% exhibited the best overall performance. Hence, it was selected for further testing of the freshness and intelligent color development effect of litchi.

### 3.8. Sensory Evaluation of Litchi Fruit

Figure 6 depicts the sensory evaluation score curve of litchi during storage, and it can be observed from Figure 6 that with the extension of storage time, the sensory evaluation scores of litchis in each group were gradually reduced. However, the sensory evaluation scores of litchis stored at 25 °C (Figure 6a) were significantly higher than those stored at 4 °C (Figure 6b). The sensory evaluation scores of the experimental group were always higher than those of the control group during storage. The control group, stored at 25 °C, exhibited the highest sensory evaluation scores. In the first two days of storage, the color of the litchi peel was bright red; the flesh was full, soft, and moderate, with a unique litchi fruit flavor, and the differences with respect to the experimental group were small. On the fourth day, the control group litchi peel turned brown, with flesh softening and turning yellow. Moreover, the taste of the fruit became slightly off-flavored with a sour taste, and the differences with respect to the experimental group increased significantly. Low-temperature storage can significantly extend the shelf life of litchi. On the 40th day of storage, the litchi peel control group was extremely moldy. A large amount of water loss resulted in a reduction in the fruit volume and the flesh texture was soft with extreme decay and deterioration. A significant odor was observed and the fruit was inedible. The experimental group of the litchi peel contained a small amount of mold. The flesh was slightly shriveled and with a slight odor; the fruit was inedible.

### 3.9. Browning Index

The browning of the fruit is primarily a disruption of the cell structure and degradation of anthocyanins in the peel, which is usually a key factor in measuring the post-harvest quality of litchi fruit. When the browning area of the litchi peel exceeds 1/2 of the peel area, it is considered to have lost its commercial value [46] (Tian et al., 2005). In this experiment, two storage temperatures were set, with a storage temperature of 25 °C, primarily used to simulate the ambient temperature of litchi stored at room temperature in most cases. A storage temperature of 4 °C was used to simulate a low-temperature storage environment. The experimental group used a double-layer antimicrobial color development material to package the litchi fruit, while the control group did not undergo any treatment. As shown in Figure 6, with an increase in storage time, the browning index of litchi fruits exhibited an increasing trend, while the browning rate of the litchi stored at 25 °C (Figure 6c) was significantly higher than that of the litchi stored at 4 °C (Figure 6d). On the third day of storage at 25 °C, the control group browning index was 0.5 and the experimental litchi browning index was 0.37. On the eighth day of storage, all fruit peels in the control group were completely brown and the browning index reached 1.00, while the experimental group’s browning index was 0.75. The storage time for both the experimental and control litchi at 4 °C was substantially extended, and the browning index of the control group remained higher than that of the experimental group throughout the process. On day 15, the browning index reached 0.36, and the browning index of the experimental group reached 0.34. After 15 days, the differences in the browning rates of the two groups gradually increased. On the 40th day of storage, the browning index of the control group reached 0.94, while the experimental group’s browning index reached 0.84. From these results, it is clear that the double-layer antibacterial color development material is beneficial for inhibiting the browning index and extending the shelf life of litchi. This may be because the double-layer antibacterial coloring material inhibits the formation of microcracks in the litchi peel and has a strong antibacterial effect, which, to a certain extent, prevents the invasion of pathogens, thereby delaying the browning of litchi.

### 3.10. Soluble Solids Content

The sweetness of litchi is primarily derived from the soluble sugars in the flesh, which also affects the flavor and quality of litchi fruit. In this experiment, the changes in the soluble solid content of litchi fruits during storage were investigated, and the results are shown in Figure 6. Freshly picked litchi was used, and the soluble solid content of the fruit on the day of picking was 17.1%. It was observed that with an increase in storage time, the soluble solid content gradually decreased. The higher the storage temperature of litchi, the faster the decrease in the soluble solids content is. At 25 °C (Figure 6e), on the fifth day of storage, the soluble solids content for the control group litchi was 16.5% compared to the day of picking, which was a reduction of 8.8%. The soluble solids content for the experimental group litchi was 17.0% compared to the day of picking, which is a reduction of 6.0%. This result indicated that the double-layer antimicrobial color development material could reduce the consumption of soluble solids in litchi during storage.

### 3.11. Titratable Acid Content

Titratable acid is the main source of acidity in litchi and has a direct effect on fruit quality and flavor. A decrease in the titratable acid content results in a decrease in fruit flavor and texture. Figure 6 depicts the change in the titratable acid content during litchi storage; the titratable acid content of litchi fruit on the first day of storage was 0.26%. The content gradually decreased with increasing storage time. The titratable acid content of litchi fruit stored at 25 °C (Figure 6g) decreased faster than that of litchi fruit stored at 4 °C (Figure 6h).

### 3.12. Litchi Freshness Intelligent Color-Rendering Effect

The freshness of litchi fruits can be directly related to color [47]. The results of the intelligent color development of the double-layer antibacterial chromogenic material for litchi fruit are shown in Figure 6 I. It can be observed from the figure that with an increase in the storage time, the total color change (ΔE) of the double-layer antibacterial chromogenic material also increased gradually. The fruit quality decreased rapidly after storage at 25 °C, whereas the ΔE value increased rapidly. On the 2nd day of storage, the ΔE value reached 4.44 ± 0.25 and the change in color could be observed by the naked eye. On the 12th day of storage, the peel had undergone severe browning and the fruit had shrunk considerably. This was accompanied by an outflow of the juice. The ΔE value of the double-layer antibacterial color-developing material reached 15.81 ± 0.41 and the color change of the material was apparent. The change in the ΔE value of the litchi stored at 4 °C was smaller than that of the litchi stored at 25 °C during the same time. This may be due to the fact that low-temperature conditions extend the shelf life of the fruit and inhibit its spoilage and deterioration, thereby reducing the color development effect of the smart material. In contrast, this may also be due to the fact that since anthocyanins are more stable at low temperatures, the reaction rate of the diphenylbenzopyran cations is slowed down, and the degradation of anthocyanin autogenesis is reduced, resulting in a lower degree of color change. A physical diagram of the intelligent color development of the double-layer antibacterial chromogenic material (before and after litchi storage) is depicted in Figure 6j. It can be observed that the color of the double-layer antibacterial chromogenic material changed with a decline in the quality of litchi. The color changed gradually from blue (at the beginning of the experiment) to red. These changes can be distinguished by the naked eye. The above results indicate that the double-layer antibacterial coloring material in the litchi storage process underwent a significant color change. This color change can be observed by the naked eye. Moreover, since the degree of change in the freshness of litchi is equivalent to the color change, the litchi freshness indication has the potential for further development.

Based on the results of the above intelligent color-rendering experiment, sensory evaluation score of the litchi fruit, and preservation-related experimental data, we can define the difference in the freshness of the litchi fruit (Table 6). The litchi fruit can be divided into five levels: fresh, fresher, not fresh, may deteriorate, and deteriorated. A detailed RGB value-based quantitative summary and results for the two-layer antibacterial intelligent color-rendering material are presented in Table 6. As shown in the table, when the freshness of the litchis was the highest, the RGB values of the double-layer antibacterial coloring material after intelligent color rendering were 145–155, 159–162, and 171–179, respectively. During the preparation and beginning of the experiment, the material exhibited a gray–blue color. With a decrease in litchi quality, the color of the material gradually changed. When the litchi changed to a non-fresh state, the RGB values of the material changed to 140–142, 139–146, and 163–169, respectively, and the material was slightly reddened. As the quality of litchi continued to decrease from the “may deteriorate” state to the “deteriorated” state, the color of the material gradually changed to purple–gray. The overall color change was significantly apparent and could be determined by the naked eye (consistent with the color change measurements in Figure 6a). It can be observed that the double-layer antibacterial color-rendering material can change with the degree of freshness of litchi. The basic realization of intelligent color rendering in real-time monitoring of the degree of freshness of litchi fruit has a significant developmental value.

## 4. Conclusions

A double-layer antibacterial chromogenic material, based on different proportions of CS and HEC, was developed by extending the flow to 3D-printing technology. The structural, physical, and functional properties of the materials were characterized. When the inner film filling rate was 45% (CHMT-45%), the physical and chemical performance of the bilayer film was the best. The diameters against *E*. *coli* and *S*. *aureus* under UV light were 10.14 mm and 11.98 mm, respectively, and they had strong color stability at varying temperatures. The results indicated that low temperatures can extend the shelf life of litchi fruit to a certain extent. The double-layer antibacterial chromogenic material determined the degree of freshness of the litchi fruit to a certain extent. In addition, with prolonged storage time, the quality of the litchi decreased, and the color of the double-layer antibacterial chromogenic material changed. Therefore, the developed material can be potentially utilized as a freshness indicator for litchi fruit.

## Figures and Tables

**Figure 1 foods-11-03286-f001:**
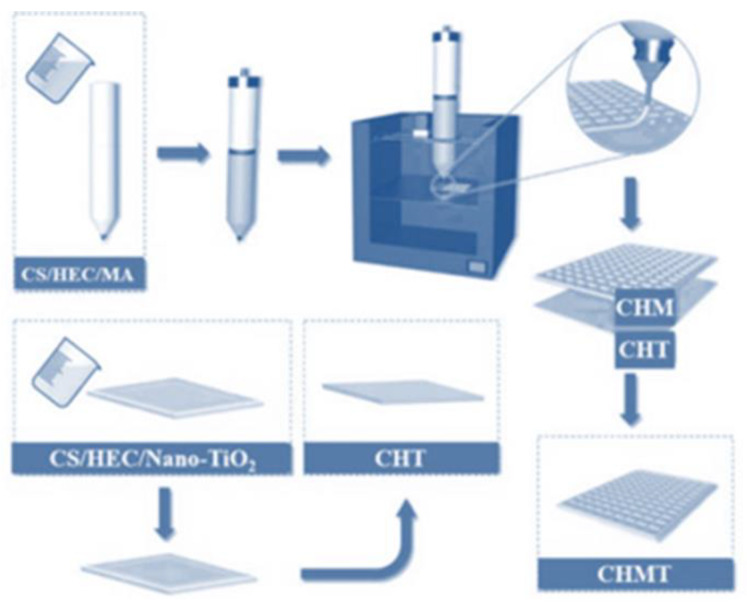
Preparation process of the 3D-printed bilayer films.

**Figure 2 foods-11-03286-f002:**
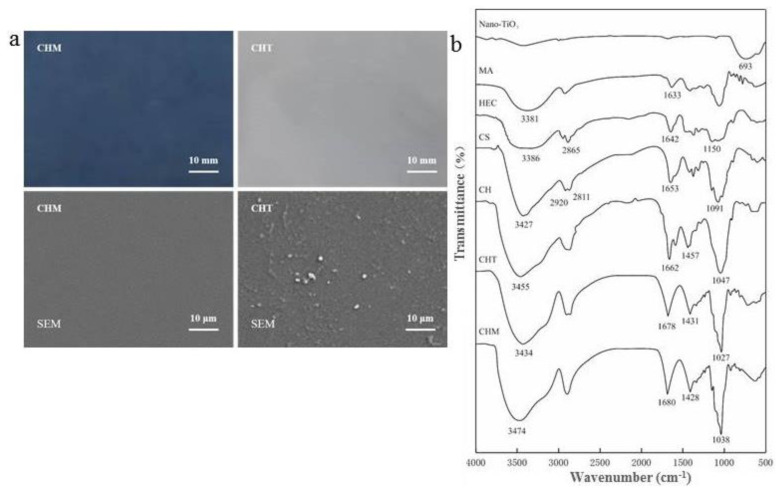
(**a**) The appearance and SEM images of the CHM and CHT materials, (**b**) Infrared spectra of each material.

**Figure 3 foods-11-03286-f003:**
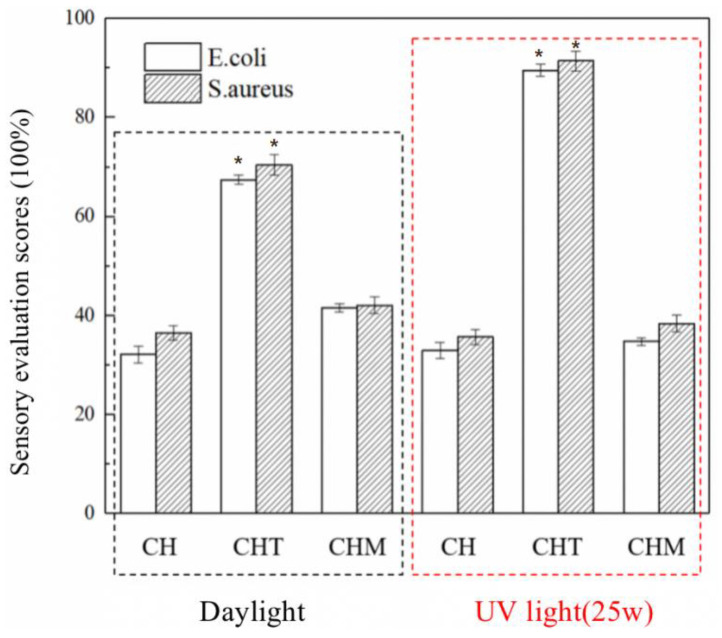
The broad-spectrum bacteriostatic rates of CH, CHT, and CHM under different light treatments. The symbol of “*” indicates that the data are significantly different (*p* < 0.05).

**Figure 4 foods-11-03286-f004:**
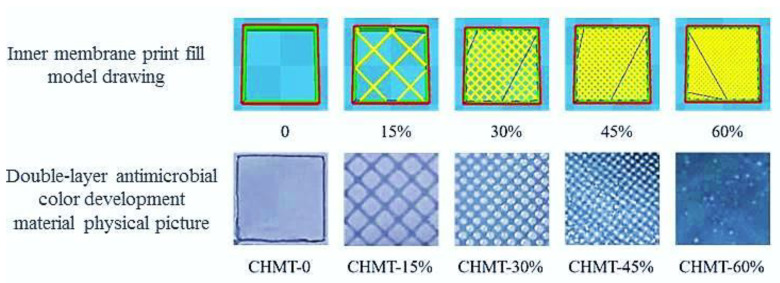
The different 3D-printing filling models of inner-layer material using CURA software, red line indicates the print boundary, yellow line indicates the gridded fill pattern; and the broad-spectrum bacteriostatic rates of CH, CHT and CHM under different light treatments.

**Figure 5 foods-11-03286-f005:**
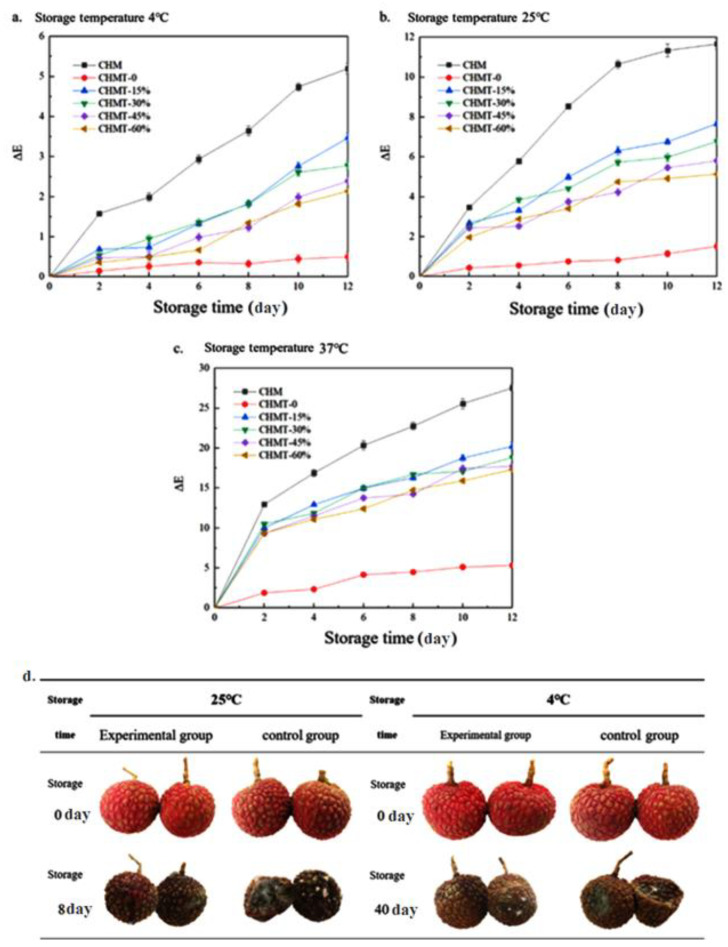
(**a**–**c**) Color stability of double-layer antibacterial chromogenic materials during storage; (**d**) Comparison of images of litchi before and after storage.

**Figure 6 foods-11-03286-f006:**
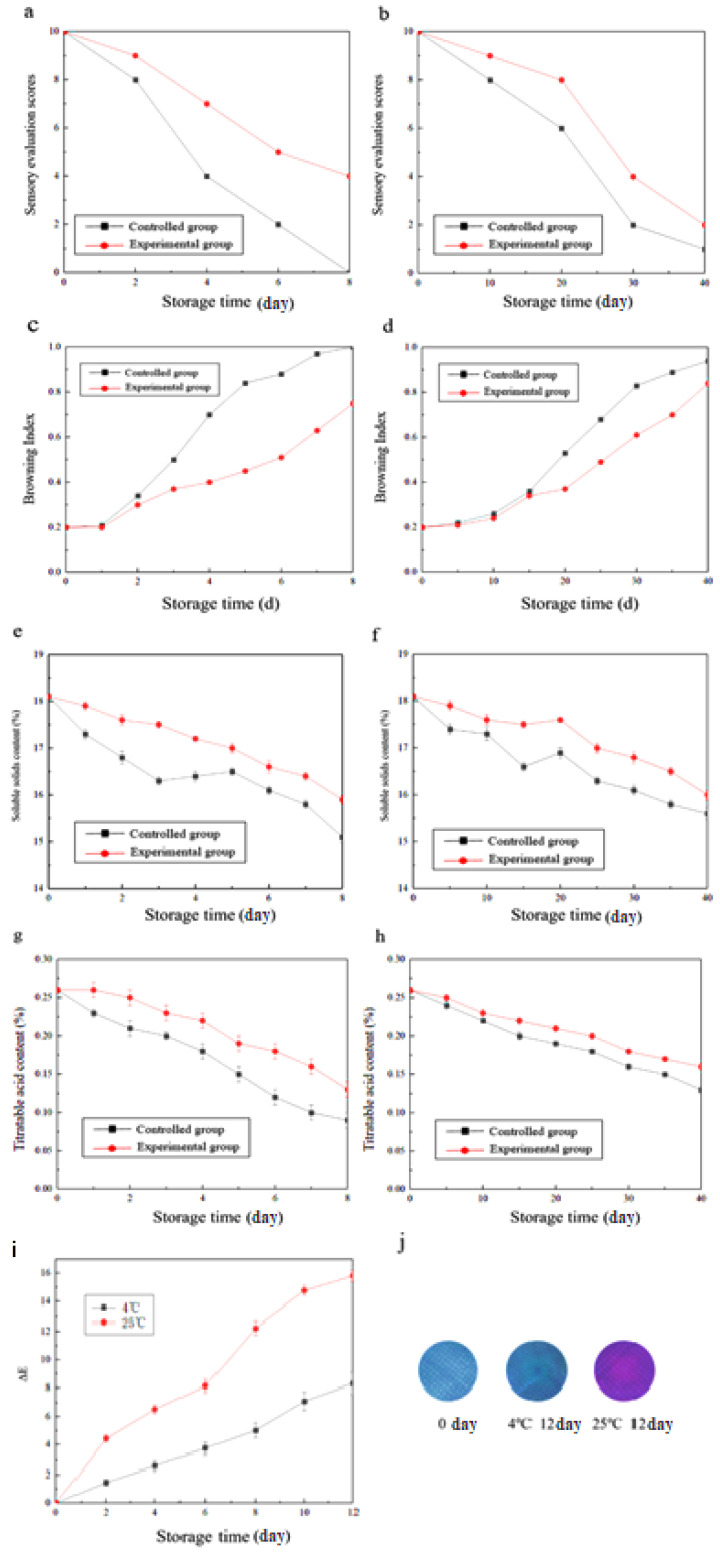
(**a**) Sensory evaluation score curve for storage temperature of 25 °C and (**b**) storage temperature of 4 °C; litchi peel browning index at (**c**) storage temperature of 25 °C and (**d**) storage temperature of 4 °C; Soluble solids content of the fruit at (**e**) storage temperature of 25 °C and (**f**) storage temperature of 4 °C; Titratable acid content of litchi fruit at (**g**) storage temperature of 25 °C and (**h**) storage temperature of 4 °C; (**i**) Variation in the chromatic aberration of double-layered antibacterial chromogenic materials in litchi during storage; (**j**) Change in the color of the double-layer antibacterial chromogenic materials before and after the intelligent color-rendering effect.

**Table 1 foods-11-03286-t001:** The materials used for the 3D-printed bilayer films.

Abbreviation	Materials
CS	Chitosan
HEC	Hydroxyethyl cellulose
MA	Mulberry anthocyanins
CH	Chitosan and hydroxyethyl cellulose composite
CHT	Chitosan, hydroxyethyl cellulose and nano-TiO_2_ composite
CHM	Chitosan, hydroxyethyl cellulose and mulberry anthocyanins composite
CHMT	Chitosan, hydroxyethyl cellulose, mulberry anthocyanins and nano-TiO_2_ composite

**Table 2 foods-11-03286-t002:** Sensory evaluation standard of litchi.

Level	State of the Skin	State of the Pulp	Smell	Taste	Score
I	Bright color, normal appearance	Full and springy	Fragrant and sweet	Taste excellent	100
II	Dark color, slight loss of water	Neither too hard, nor too soft	Natural fragrance	Taste good	80–90
III	Large area browning and water loss	Water softening	Light herbal scent	Poor taste	60–70
IV	Complete browning and large loss of water	The flesh is yellow and shriveled	Slight abnormal taste	Taste sour	40–50
V	Mildew spots and severe water loss appear	Atrophy and slight decay	Abnormal taste	Almost inedible	20–30
VI	Severe mildew and water loss	Putrid	Obvious abnormal taste	Cannot eat	0–10

**Table 3 foods-11-03286-t003:** Chroma and opacity of the double-layer antibacterial chromogenic materials.

Sample	*L**	*a**	*b**	Δ*E*	Opacity
CHMT-0	88.78 ± 0.43 ^a^	0.77 ± 0.01 ^a^	0.41 ± 0.07 ^a^	4.82 ± 0.19 ^a^	21.53 ± 0.04 ^a^
CHMT-15%	80.39 ± 0.79 ^b^	11.02 ± 0.77 ^b^	−4.66 ± 0.34 ^b^	12.86 ± 0.20 ^b^	22.03 ± 0.09 ^a^
CHMT-30%	72.64 ± 0.53 ^c^	13.84 ± 0.67 ^c^	−6.33 ± 0.42 ^c^	20.34 ± 0.02 ^c^	33.55 ± 0.56 ^b^
CHMT-45%	69.69 ± 1.01 ^d^	15.53 ± 0.32 ^d^	−7.35 ± 0.42 ^d^	23.78 ± 0.89 ^d^	39.40 ± 0.41 ^c^
CHMT-60%	63.56 ± 0.89 ^e^	16.33 ± 0.24 ^d^	−8.69 ± 0.31 ^e^	29.36 ± 0.93 ^e^	46.34 ± 0.47 ^d^

Note: The results are expressed as standard deviations and ± averages; different letters after each column indicate significant differences (*p <* 0.05).

**Table 4 foods-11-03286-t004:** Thickness, water vapor transmission coefficient, and mechanical properties of the double-layer antibacterial chromogenic materials.

Sample	Thickness (μm)	WVP·10^−12^ (g·cm/(cm^2^·s·Pa))	Tensile Strength (Mpa)	Elongation at Break (%)
CHMT-0	40.01 ± 0.28 ^a^	7.81 ± 0.25 ^a^	21.55 ± 0.58 ^a^	17.01 ± 0.11 ^a^
CHMT-15%	63.24 ± 1.22 ^b^	7.23 ± 0.53 ^a^	22.32 ± 0.65 ^a^	18.01 ± 1.18 ^a^
CHMT-30%	70.12 ± 1.34 ^c^	6.54 ± 0.33 ^b^	25.61 ± 0.39 ^b^	20.01 ± 0.25 ^b^
CHMT-45%	79.81 ± 1.76 ^d^	4.73 ± 0.45 ^c^	29.52 ± 0.77 ^c^	21.01 ± 0.32 ^b^
CHMT-60%	87.21 ± 1.53 ^e^	2.16 ± 0.15 ^d^	34.43 ± 1.12 ^d^	24.01 ± 0.73 ^c^

Note: Different letters after each column indicate that the data are significantly different (*p <* 0.05).

**Table 5 foods-11-03286-t005:** Size of the inhibition zone of varying double-layer antibacterial chromogenic materials under varying light conditions.

Sample	Inhibitory Circle Size in Daylight (mm)		Post-UV Inhibition Circle Size (mm)	
	*E. coli*	*S. aureus*	*E. coli*	*S. aureus*
CHMT-0	14.58 ± 0.11 ^a^	16.44 ± 0.13 ^a^	16.12 ± 0.19 ^a^	18.96 ± 0.24 ^a^
CHMT-15%	13.43 ± 0.15 ^b^	15.38 ± 0.16 ^b^	15.07 ± 0.10 ^b^	17.52 ± 0.17 ^b^
CHMT-30%	11.26 ± 0.20 ^c^	14.11 ± 0.09 ^c^	12.15 ± 0.23 ^c^	15.67 ± 0.19 ^c^
CHMT-45%	9.33 ± 0.08 ^d^	11.19 ± 0.17 ^d^	10.14 ± 0.12 ^d^	11.98 ± 0.25 ^d^
CHMT-60%	8.24 ± 0.09 ^e^	9.27 ± 0.05 ^e^	8.43 ± 0.04 ^e^	9.66 ± 0.12 ^e^

Note: Different letters after each column indicate that the data are significantly different (*p <* 0.05).

**Table 6 foods-11-03286-t006:** Corresponding RGB values of the intelligent color rendering of litchi with varying degrees of freshness by double-layer antibacterial chromogenic materials.

	Freshness of Litchi				
	Fresh	Relatively Fresh	Stale	Almost Rot	Spoilage
Sensory evaluation score	100	80–90	60–70	40–50	0–30
Red (R)	145–155	124–132	140–142	150–159	155–163
Green (G)	159–162	144–156	139–146	130–141	122–131
Blue (B)	171–179	162–170	163–169	167–175	168–173
Freshness colorimetric card					

## Data Availability

Data is contained within the article.

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
