# Peer review of "Fabrication of Chitosan/Hydroxyethyl Cellulose/TiO2 Incorporated Mulberry Anthocyanin 3D-Printed Bilayer Films for Quality of Litchis"

_foods, 2022, doi:10.3390/foods11203286_

Round 1
Reviewer 1 Report
L12 Which form of 3D printing? Ink?
L91 Objective of this research should be clearly stated here. The manuscript contains large amounts of works which can make reader confusing.
L311 Add more discussion e.g., There was shifting of the wide peak between 3800-3000 cm-1 which suggested hydrogen bonding between polymer components including interaction between polymers and TiO2 via hydrogen bonding (doi.org/10.3390/polym13234192).
L311-312 Only 3000 -2750 attributes to C-H stretching. The 3750-3000 cm-1 comes from O-H, N-H stretching. “3750-3000” should be removed.
Fig. 3 What are these different colors refer to? What are the pattern in these photos? The caption should briefly explain.
L360 Add more discussion e.g., Denser matrices with few pores or void spaces leads to lower permeability of the polymeric matrices (doi.org/10.1016/j.fpsl.2021.100787).
L369 and others -> Recheck error in citation
L380 Add more discussion e.g., The filling improved homogeneity of the polymeric matrices possibly by filling up the void spaces, giving better distribution of stress due to applied force and improved mechanical properties (doi.org/10.1016/j.foodchem.2021.131709).
Fig. 4 Add statistical analysis
Fig. 5 Recheck spelling “temperature”
What are experimental vs controlled group? What are different between litchi in different groups as they look all the same.
Section 3.8-3.12 are confusing. How these parts refer to previous sections should be clearly stated. Which samples from 3-D materials are selected for these experiments should be stated in both materials and method and results and discussion section.
There should also be discussion on comparison between groups in all section
Conclusions should be carefully revised as there are two parts of research.
Author Response
Comments 1:
L12 Which form of 3D printing? Ink?
Re: 3D printing ink, thanks.
L91 Objective of this research should be clearly stated here. The manuscript contains large amounts of works which can make reader confusing.
Re: This part was rewritten, thanks.
L311 Add more discussion e.g., There was shifting of the wide peak between 3800-3000 cm-1 which suggested hydrogen bonding between polymer components including interaction between polymers and TiO2 via hydrogen bonding (doi.org/10.3390/polym13234192).
Re: Added, thanks.
L311-312 Only 3000 -2750 attributes to C-H stretching. The 3750-3000 cm-1 comes from O-H, N-H stretching. “3750-3000” should be removed.
Re: Deleted, thanks.
Fig. 3 What are these different colors refer to? What are the pattern in these photos? The caption should briefly explain.
Re: Fig. 3 (a) The different 3D printing filling models of inner-layer material using CURA software, red line indicate the print boundary, yellow line indicate the gridded fill pattern, thanks.
L360 Add more discussion e.g., Denser matrices with few pores or void spaces leads to lower permeability of the polymeric matrices (doi.org/10.1016/j.fpsl.2021.100787).
Re: Added, thanks.
L369 and others -> Recheck error in citation
Re: We corrected the wrong description, thanks.
L380 Add more discussion e.g., The filling improved homogeneity of the polymeric matrices possibly by filling up the void spaces, giving better distribution of stress due to applied force and improved mechanical properties (doi.org/10.1016/j.foodchem.2021.131709).
Re: Added, thanks.
Fig. 4 Add statistical analysis
Re: Added, thanks.
Fig. 5 Recheck spelling “temperature”
Re: Corrected, thanks.
What are experimental vs controlled group? What are different between litchi in different groups as they look all the same.
Re: It was the first day after picking, so the color of the litchi in the experimental and control groups was the same, and we replaced the Fig. 5, thanks.
Section 3.8-3.12 are confusing. How these parts refer to previous sections should be clearly stated. Which samples from 3-D materials are selected for these experiments should be stated in both materials and method and results and discussion section.
Re: Yes, you are right, we chose CHMT-45% for further experiment on fresh-keeping and color development of Litchi, thanks.
There should also be discussion on comparison between groups in all section
Re: We use CHMT-45% instead of experimental group. Because CHMT-45% has been proved have the optimal physical and chemical properties in the early stage of the experiment, we need to verify the accuracy of the previous ones in the application and color development experiment, so we only investigated CHMT-45% group when it was applied to litchi, thanks.
Conclusions should be carefully revised as there are two parts of research.
Re: The conclusion has been rewritten, thanks.
Reviewer 2 Report
I am sorry to inform that the authors should revise their paper significantly. The authors have developed five various membranes using different ratios of chitosan and hydroxyethyl cellulose and with the addition of nano-TiO2. However, it is not clear what kind of materials are CHM and CHT. Then in Figure 1, the authors have shown FT-IR spectra of various materials: nano-TiO2, MA, HEC, CS, CH, CHT, and finally CHM. Figure 3 and Table 2 shows properties of CHMT-0, CHMT-15%, CHMT-30% etc. I do not know what kind of materials are materials with abbreviation CHMT. The authors should summarize the materials they have developed in a table to clarify the abbreviation of the membranes, their composition, method of execution, etc.
Some other remarks:
What is the difference between sections 2.8 and 2.11? The methods are the same; please link these two points.
Figure 3 and Figure 5 should be provided in a better quality
Author Response
Comments 2:
I am sorry to inform that the authors should revise their paper significantly. The authors have developed five various membranes using different ratios of chitosan and hydroxyethyl cellulose and with the addition of nano-TiO2. However, it is not clear what kind of materials are CHM and CHT. Then in Figure 1, the authors have shown FT-IR spectra of various materials: nano-TiO2, MA, HEC, CS, CH, CHT, and finally CHM. Figure 3 and Table 2 shows properties of CHMT-0, CHMT-15%, CHMT-30% etc. I do not know what kind of materials are materials with abbreviation CHMT. The authors should summarize the materials they have developed in a table to clarify the abbreviation of the membranes, their composition, method of execution, etc.
Re: We have re-summarized the abbreviations in the table 1 to make it easily for readers, thanks.
Some other remarks: What is the difference between sections 2.8 and 2.11? The methods are the same; please link these two points.
Re: Yes, we deleted the sections 2.8, thanks.
Figure 3 and Figure 5 should be provided in a better quality
Re: Figure 3 and Figure 5 were replaced, thanks.
Round 2
Reviewer 1 Report
The manuscrupt has been improved.
Reviewer 2 Report
The authors have improved their work, the paper can be accepted in this form.